# Efficacy of Plantar Orthoses in Paediatric Flexible Flatfoot: A Five-Year Systematic Review

**DOI:** 10.3390/children10020371

**Published:** 2023-02-13

**Authors:** Cristina Molina-García, George Banwell, Raquel Rodríguez-Blanque, Juan Carlos Sánchez-García, Andrés Reinoso-Cobo, Jonathan Cortés-Martín, Laura Ramos-Petersen

**Affiliations:** 1Health Sciences Ph.D. Program, Universidad Católica de Murcia UCAM, Campus de Los Jerónimos n°135, Guadalupe, 30107 Murcia, Spain; 2Department of Nursing and Podiatry, Faculty of Health Sciences, University of Malaga, Arquitecto Francisco Peñalosa 3, Ampliación de Campus de Teatinos, 29071 Malaga, Spain; 3Research Group CTS1068, Andalusia Research Plan, Junta de Andalucía, Nursing Department, Faculty of Health Sciences, University of Granada, 18071 Granada, Spain; 4San Cecilio Clinical University Hospital, 18016 Granada, Spain

**Keywords:** flatfoot, paediatrics, child, foot orthoses

## Abstract

Paediatric flexible flatfoot (PFF) is a very common condition and a common concern among parents and various healthcare professionals. There is a multitude of conservative and surgical treatments, with foot orthoses (FO) being the first line of treatment due to their lack of contraindications and because the active participation of the child is not required, although the evidence supporting them is weak. It is not clear what the effect of FO is, nor when it is advisable to recommend them. PFF, if left untreated or uncorrected, could eventually cause problems in the foot itself or adjacent structures. It was necessary to update the existing information on the efficacy of FO as a conservative treatment for the reduction in signs and symptoms in patients with PFF, to know the best type of FO and the minimum time of use and to identify the diagnostic techniques most commonly used for PFF and the definition of PFF. A systematic review was carried out in the databases PubMed, EBSCO, Web of Science, Cochrane, SCOPUS and PEDro using the following strategy: randomised controlled trials (RCTs) and controlled clinical trials (CCTs) on child patients with PFF, compared to those treated with FO or not being treated, assessing the improvement of signs and symptoms of PFF. Studies in which subjects had neurological or systemic disease or had undergone surgery were excluded. Two of the authors independently assessed study quality. PRISMA guidelines were followed, and the systematic review was registered in PROSPERO: CRD42021240163. Of the 237 initial studies considered, 7 RCTs and CCTs published between 2017 and 2022 met the inclusion criteria, representing 679 participants with PFF aged 3–14 years. The interventions of the included studies differed in diagnostic criteria, types of FO and duration of treatment, among others. All articles conclude that FO are beneficial, although the results must be taken with caution due to the risk of bias of the included articles. There is evidence for the efficacy of FO as a treatment for PFF signs and symptoms. There is no treatment algorithm. There is no clear definition for PFF. There is no ideal type of FO, although all have in common the incorporation of a large internal longitudinal arch.

## 1. Introduction

Paediatric flexible flatfoot (PFF) is a common condition in children [1,2]. Ninety percent of appointments in foot clinics are related to flat feet (FF) [3]. Epidemiological studies indicate that 4% of 10-year-old children suffer from PFF and 10% of them are under treatment to prevent secondary pathologies during adulthood [4]. In addition, PFF is a common concern for parents and a highly debated topic by all healthcare professionals [5,6,7].

For decades, up to the present, PFF has been a highly controversial issue, being difficult to differentiate what is normal or pathological. There are also questions about how to diagnose it, when it should be treated, when the physiological evolution should be allowed to continue, what is the best conservative treatment and when surgical treatment is necessary [8,9,10].

There is no universally accepted or precise definition for FF. Clinically, FF is understood as a flattening of the medial longitudinal arch when the subject is in a standing position [11]. FF is a triplanar presentation of the foot [12], accompanied by a valgus position of the calcaneus, medial prominence of the talus, flattened footprint, abduction of the forefoot with respect to the hindfoot and internal rotation of the tibia [5,13]. In the PFF, it is possible to correct the deformity when the person is not in a standing position, where the arch is present [14]. Therefore, there is controversy among different professionals regarding its treatment. Some professionals indicate that it is a physiological variant of foot development and that it will correct itself in time [15]. Other professionals indicate that PFF will slowly lead to pathologies in the foot, ankle or proximal structures, such as plantar fasciopathy, Achilles and posterior tibial tendinopathy, hallux limitus and rigidus, chondromalacia patellae and patellofemoral pain syndrome [5,16,17,18,19,20].

The most common symptoms are functional disability and general foot and leg pain, although the majority of PFF cases are asymptomatic [21]. Regardless of symptomatology, there are biomechanical abnormalities, including decreased ankle dorsiflexion, increased hindfoot eversion and forefoot supination [5,13,22].

The diagnosis is based on clinical tests, analysis of the footprint or radiology. The most widely used clinical tests are relaxed calcaneus position in standing, neutral calcaneal position in standing position, navicular drop, navicular drift, navicular height, foot posture index (FPI), Jack’s test, double/single heel rise test (HRT), maximum pronation test, supination resistance test, pronation angle and too many toes test. Furthermore, assessments and evaluations such as genu valgus, asymmetry, tibial torsions, metatarsus adductus, flexibility assessment (most commonly assessed by Beighton scale or the Wynne-Davies criteria [23]), etc.) are used. In terms of the analysis of the footprint, it can be conducted by pedigraph, pressure platform, photopodogram and podoscope. Finally, radiographic measurements are the most objective ones, including assessment from two load projections. The most common radiology assessments are lateral and dorsoplantar talocalcaneal angle, angle of inclination of the calcaneus, talus and first metatarsal, medial and lateral Costa-Bartani angle, talus–first metatarsal angle, calcaneus–fifth metatarsal, tibial talus, line of Cyma and Schade [5,8,24,25,26,27,28].

There is a wide range of treatments for PFF. Evidence is lacking or very limited for most conservative treatments. Since there are no established criteria to differentiate a pathological PFF from a physiological one, the decision to treat PFF depends on each clinician [13]. The factors that are considered when establishing a treatment are age, flexibility, severity of the deformity, equinus position, adequate footwear and the presented symptoms [29]. Surgery, including procedures such as subtalar arthroereisis [30], is reserved for feet that have severe deformity, rigid FF or FF with persistent symptoms that do not improve with conservative treatment. The most used conservative treatments are foot orthoses (FO), corrective shoes, physical exercises, physiotherapy with joint manipulations and the Mulligan method [5,7,13,16,19,31,32,33,34]. The most frequent conservative intervention is the use of FO [25,35].

The short-term purpose of treatment with FO is to decrease pronator movement, hence decreasing the tensile forces on ligaments, tendons and the plantar fascia. The long-term goal would be to reduce the pathological position of the foot and slow down the progression [22,29,36,37]. FO treatment has been modified and has evolved over time, including thermoplastic, polypropylene FO and postings which aim to achieve a neutral hindfoot position [38]. The current evidence of FO treatments is very limited as systematic reviews have demonstrated; some of them concluded that FO present efficacy and some of them did not [13,14,21,29,31,35,39,40]. Recent studies continue to show ambiguity, although the evidence on the efficacy of FO is increasing, especially when FO are customised [4,19,21,41,42,43,44,45,46,47,48].

Recent studies conclude that PFF should not be ignored, and their treatment should not be downplayed, considering that the sooner that effective treatment is prescribed, the less damage will occur to other parts of the body. They also add that a conservative corrective treatment should be carried out, rather than an invasive treatment [19,20].

Therefore, since untreated PFF could cause problems in the foot itself or in other structures, it is necessary to demonstrate the efficacy of FO as a conservative treatment to reduce signs and symptoms in patients with PFF. It is also important to know the best type of FO and the minimum time of use as well as to identify which are the most used diagnostic techniques for PFF and how it is defined.

## 2. Materials and Methods

This protocol was registered on the International Prospective Register of Systematic Reviews PROSPERO: CRD42021240163. In order to respond to the objectives set out in this present study, a systematic review was carried out following the regulations “Preferred reported items of systematic reviews and meta-analysis” (PRISMA) and in accordance with the recommendations of the Cochrane Collaboration [49].

### 2.1. Selection Criteria

#### 2.1.1. Types of Studies

Randomised controlled trials (RCTs) and controlled clinical trials (CCTs) published in the last five years were included. All other types of studies, such as systematic reviews, were excluded.

#### 2.1.2. Participants

Studies included in this review had to include children diagnosed with PFF. Patients who had surgery in the lower limbs, or who presented some systemic or infectious neurological disease were excluded.

#### 2.1.3. Type of Intervention

Interventions which were considered included FO as treatments for at least 2 months, both customised or prefabricated, and/or with modifications.

#### 2.1.4. Comparison

Studies that compared the intervention with another type of FO or placebo.

#### 2.1.5. Outcome Measure

The outcomes considered were those used to evaluate the improvement of signs and symptoms of the PFF.

### 2.2. Search Strategy

The search was carried out by two researchers independently in the following databases: PubMed, EBSCO, Web of Science, Cochrane, SCOPUS and PEDro. The following medical subject headings (MeSH) were used: flatfoot, paediatrics, child, foot orthoses, according to the characteristics of each database, accompanied by the Boolean operators “AND” and “OR”.

The following search strategy was used: ((“Flatfoot”[Mesh] AND (“Paediatrics”[Mesh] OR “Child”[Mesh] OR “Child, Preschool”[Mesh]) AND “Foot Orthoses”[Mesh]) OR ((“Flexible Flatf**t”[tw] OR “Flat F**t”[tw] OR “Pes Planus”[tw] OR Flatf**t[tw] OR Splayfoot[tw] OR “F**t, Flat” [tw] OR “Flatf**t, Flexible”[tw]) AND (Paediatrics[tw] OR “Preschool Child*”[tw] OR Child*[tw] OR “Child*, Preschool”[tw]) AND (“Foot Orthosis”[tw] OR “Orthotic Insole*”[tw] OR “Orthos*s, Foot”[tw] OR “Foot Orthotic Device*”[tw] OR “Device*, Foot Orthotic”[tw] OR “Orthotic Device*, Foot”[tw] OR “Arch Support*, Foot”[tw] OR “Foot Arch Support*”[tw] OR “Support*, Foot Arch”[tw] OR “Orthotic Shoe Inserts”[tw] OR “Insole*, Orthotic”[tw] OR “Orthotic Insole*”[tw])))

In addition, the papers bibliographies were reviewed.

### 2.3. Study Selection

The selection of the studies was carried out by two researchers. After the selection of the papers from the databases, the duplicates were eliminated. After the elimination, a screening of the titles and abstracts was carried out, based on the inclusion and exclusion criteria. The selected studies were then fully read to assess compliance with the eligibility criteria. Any disagreements between reviewers in any phase of study selection were resolved by consulting another reviewer.

### 2.4. Data Extraction and Management

In order to respond to the proposed objectives, data were extracted from the studies, including characteristics of the publication (author, country, year and journal of publication, study design, objectives, keywords), characteristics of the sample (sample size, age, sex, height, weight, body mass index (BMI), whether the PFF was symptomatic or not, previous treatments and diagnosis), characteristics of the diagnosis and characteristics of the intervention (FO type and material, FO use, what health education/recommendations each participant received and the duration of the treatment) and the results together with the final conclusions of each study.

### 2.5. Risk of Bias and Quality Assessment

To estimate the methodological quality/risk of bias of each of the included studies, two different types of scales were used for the two study types (i.e., RCT or CCT).

To evaluate the RCTs, the tool recommended by the Cochrane manual was used to assess the risk of bias. It is a domain-based assessment that evaluates each domain with three possibilities: ‘low risk of bias’, ‘high risk of bias’ or ‘unclear risk’ [50]. To assess the CCTs, the “Before-After Quality Assessment Tool (BAQA)” developed by the National Institute of Health (NIH) in collaboration with the Cochrane team, among others, was used. It is a tool that answers 12 very specific questions to assess key concepts of the internal validity of the studies [51].

In addition, the Scottish Intercollegiate Guidelines Network (SIGN) scale was used to reflect the level of evidence and degree of recommendation [52].

### 2.6. Data Synthesis

Data have been presented in tables and narrative forms to describe the characteristics of the included studies. As the studies were not sufficiently homogeneous, it was impossible to perform a meta-analysis.

## 3. Results

Using the search strategy outlined above, we identified a total of 237 studies in the databases, as well as 4 additional records identified through other sources, which was via the reference lists of the initial papers that were retrieved. Of these 241 items, 181 were duplicated records. The remaining 60 studies were evaluated by title and abstract by 2 independent reviewers. Of these, 19 were excluded due to differences in inclusion criteria as they were observational studies, clinical trials without control group or the participants were not children. After that, 41 full texts were assessed for eligibility and 34 were excluded because participants had previously undergone surgery in the lower limbs, or the FO they were prescribed had been used for less than 2 months, among other reasons. Thus, only seven papers fully met the inclusion criteria. Figure 1 shows the PRISMA flow diagram for the studies included in this review.

### 3.1. General Characteristics of the Studies Assessed

The included studies were published in the last 5 years and all of them were developed on the Asian continent, except for Rusu et al. [53]. Regarding the levels of evidence assessed by the SIGN grading system, the included studies presented levels II+A and II+B.

Of the 679 included participants, 412 were boys and 267 were girls, between 3 and 14 years of age. The study with the largest sample size was carried out by Chen et al. [42], with a total of 466 subjects, being more than two thirds of the total population included in this review.

In terms of the characteristics of the participants, their BMI was from 16.04 [42] to 20.1 [4,54]. Some of the studies did not provide BMI data. All the participants, to be selected, presented with PFF. Furthermore, in some of the included studies the participants presented pain [42,43,55]. None of the participants received previous treatment (Table 1).

Participants were excluded if they had some type of foot or lower limb surgery, neurological, neuromuscular or hereditary disease, or if they had a developmental or coordination/mobility disorder.

Each author defined PFF with different characteristics. To do this, all authors used clinical diagnostic tests, even Rusu et al. [53] who did not specify the diagnoses but indicated that participants were evaluated using static and dynamic assessment. In addition, only three of the included studies, Choi et al. [43], Hsieh et al. [55] and Ahn et al. [56], also used radiographic examination for diagnosis.

All the studies that performed a radiographic diagnosis evaluated the lateral and anteroposterior projections of the feet in loadbearing. In addition, Choi et al. [43] evaluated the posterior projection, which was introduced by Saltzman and el-Khoury. The angles measured in the different projections were highly variable; only three authors agreed on the same angle: the angle of inclination of the calcaneus in the lateral projection. For Hsieh et al. [55], patients were diagnosed with PFF when two of the three angles they measured were not within normal values. For Ahn et al. [56], patients were diagnosed with PFF when a radiological finding in any of the four angles that were evaluated were not within normal values. Choi et al. [43] did not specify when patients were diagnosed with PFF.

Regarding diagnostic clinical tests, eight different tests were identified: (1) navicular drop, (2) RCPS, (3) arch height index, (4) pedigraphy Chippaux–Smirak index (CSI), (5) Beighton scale, (6) Jack’s test, (7) double/single heel rise test and (8) FPI. Each author used different tests in their studies. If tests were positive, it meant that they presented PFF. For this, the navicular drop had to be greater than 10 mm for Jafarnezhadgero et al. [4,54] and greater or equal to 6 mm for Hsieh et al. [55]. Additionally, PRCA had to be greater than 4° of eversion, the arch height index less than 0.31, the CSI greater than 62.7%, the Beighton scale greater than 4 and finally the FPI had to be greater than 6 in the total score (Table 2).

#### Risk of Bias Assessment

All the included studies presented a high risk of bias in at least one field. For Ahn et al. [56] most of the items presented unclear risk of bias and the study by Rusu et al. [53] did not report whether or not the investigators were blinded or not (Figure 2).

### 3.2. Results by Outcome Measures

As required by one of our inclusion criteria, all the studies divided the sample into two different groups, an intervention group (IG) and a control group (CG). It should be added that all the authors, except Ahn et al. [56], provided FO treatment to the IG and the CG received a placebo treatment.

All the authors casted the patients’ feet, apart from Hsieh et al. [55] who provided a FO with a direct adaptation technique, and Chen et al. [42] who provided off-the-shelf FO. Choi et al. [43] used a phenolic foam, Jafarnezhadgero et al. [4,54] used plaster with the foot in a neutral position and Rusu et al. [53] used a 3D scanner.

Some authors included information about the casting of the foot and more detailed information about the FO. Rusu et al. (2022) [53] included personalised semi-rigid FO, with increased medial longitudinal arch support and heel cup, which were manufactured by the company Ortoprotesica. The design of the FO was computerised using a CAD-CAM system. In the studies of Jafarnezhadgero et al. (2020) [4] and Jafarnezhadgero et al. (2018) [54] the FO was made from ethylene-vinyl acetate (EVA) and microcellular rubber, and the negative cast was made in a subtalar joint neutral position. Chen et al. (2019) [42] included prefabricated FO, which were adapted by an orthotist. Choi et al. (2019) [43] used phenolic foam to cast the foot in a weight-bearing position. The FO were personalised with an increased medial longitudinal arch support. In the study of Hsieh et al. (2018) [55] the FO were directly adapted to participants’ feet, in an offloading and neutral position. The FO were personalised with a medial longitudinal arch support. Finally, in the study of Ahn et al. (2017) [56], the neutral position of weightbearing plaster cast technique was used to capture foot shape, and the FO manufactured were Blake´s inverted orthoses.

All the FO from the IG of the included studies had a marked medial longitudinal arch and EVA was used for some parts of the FO. However, all the FO were different. Rusu et al. [53] made semi-rigid custom FO, using a thermoplastic heel cup which extended to the base of the metatarsals and had an EVA top cover and a metatarsal dome. For the IG Jafarnezhadgero et al. [4,54] provided a resin FO with a maximum of 25 mm medial longitudinal arch, and the CG received a flat polyester resin FO. The following papers provided an intervention for the IG, but nothing for the CG. Chen et al. [42] provided polypropylene and EVA off-the-shelf FO. Choi et al. [43], provided personalised FO, including materials with different hardness, such as EVA, plastazote, poron, evazote or ucolite. Hsieh et al. [55] provided customised FO made of thermoplastic, a medial longitudinal arch made of EVA and hindfoot posting. Finally, Ahn et al. [56], provided a Blake inverted FO together with a medial longitudinal arch for the IG and a Blake inverted FO without a medial longitudinal arch for the CG.

All the authors recommended a daily use of the FO during the daily activities of life, from 3 months to 6 years. Regarding the follow-up of the participants, all the studies carried out an initial assessment (pre-treatment) and then a final assessment (post-treatment). Only Choi et al. [43], carried out an assessment every 6 months until the end of the treatment.

Different outcome measurements were evaluated in the included studies even though the objective in all the studies was the same, which was to determine the efficacy of the FO. Some of the studies evaluated radiographic changes, others evaluated kinetic-kinematics changes and others evaluated changes in the plantar footprint. Some authors highlighted functional changes and others evaluated morphological changes after FO use.

The authors used different devices and tests to quantify the results: the VICON system, pressure platform, pedigraph (CSI), radiographs, International Classification of Functionality, RCPS and static and dynamic changes (Table 3).

All authors conclude that FO are an effective treatment, although more evidence is needed to fully confirm this statement. Choi et al. [43] concluded that FO may make structural changes, and that FO improve functionality and pain. Rusu et al. [53] concluded that exercise is beneficial, particularly when combined with FO treatment. They also reported that a decrease in subtalar joint flexibility could lead to an increase in the plantar arch index. Jafarnezhadgero et al. [4,54] assessed the changes in kinetics and kinematics measured by the VICON system and pressure platform in two papers. They noted a difference in the kinetics and kinematics, concluding that the long-term use of FO with medial longitudinal arch support help to improve the alignment of the lower limbs and gait in PFF. Chen et al. [42] concluded that although PFF may resolve with age, the use of FO may reduce the characteristic signs, especially in 5-year-olds (more than in 3-year-olds). Hsieh et al. [55] concluded that FO provided a reduction in pain and an increase in comfort. Finally, the authors Ahn et al. [56] observed clinical and radiological improvements in both groups in their study, but that the IG obtained greater changes. Therefore, they concluded that a Blake inverted FO together with a medial longitudinal arch was more effective than a Blake inverted FO without a medial longitudinal arch.

## 4. Discussion

The aim of this review was to demonstrate the efficacy of FO as a conservative treatment to reduce signs and symptoms in patients with PFF. In addition, it was important to determine the best type of FO and the minimum time of use and finally, to identify which are the most used diagnostic techniques for PFF and how it is defined.

To answer the main objective, in five of the included studies [4,42,43,54,55] the CG received no treatment or placebo. This makes the point that no therapy was applied, meaning that all the outcome measures that were improved in the IC were because of the FO, not because of the natural evolution of the PFF.

This study shows that FO were an effective treatment for PFF. Recently, more studies have been published supporting the efficacy of FO, showing their positive impact on a wide variety of PFF outcomes such as pain, foot posture, gait, foot function, etc. [35]. This review shows a different perspective from previously published research where a positive impact from the FO was not shown.

However, no ideal type of FO has been agreed in the literature. Each author used a different FO, including different types and materials, though always rigid or semirigid materials. However, all the FO had something in common, which was a high longitudinal medial arch support. A recent study has shown that the use of custom FO for PFF is more effective than prefabricated FO, providing better pressure distribution and conform better to the foot [44]. Su et al. [57] concluded that there is a relationship between hardness of the FO and effectiveness of treatment, however the increase in hardness was also linked to soft tissue damage.

In terms of FO use, all included studies specified that FO should be worn every day, the period of which varied from 3 months to 6 years. Reviewing the literature, there is no consensus on how long children with PFF should wear their FO, with differing periods from 3 months to 2 years. However, some authors consider 3 months an insufficient amount of time [21,38]. Radwan et al. [45] concluded that FO can modify children’s feet with immediate effect, but it is after 12 months when more changes and improvements are shown. Jafarnezhadgero et al. [4,54] concluded that long-term FO use was effective to improve alignment and coordination of the lower limbs, as well as gait kinetics and kinematics. Those results agree with previous studies [45,58]. Chen et al. [42] and Hsieh et al. [55] also concluded that FO were effective, reducing the characteristic signs and symptoms of PFF and improving quality of life, which agrees with previous studies [54,59,60,61,62,63]. However, none of the included studies indicated negative effects from the use of FO; previous studies indicated localised irritation of the skin, increased pain, problems with shoe fit, intolerance or discomfort after FO use in some of the participants [46].

Previous studies concluded that the use of footwear is part of the treatment in order to ensure the effectiveness of the FO [25,38]. However, only Jafarnezhadgero et al. [4,54] recommended a specific type of footwear for the participants.

Age may be the characteristic that most influenced the results and the evolution of the treatment [25,64]. The mean age of most of the included participants was 10 (except for the studies by Chen et al. [42] and Hsieh et al. [55]). Depending on the study, the ideal age to treat PFF varies. Some studies concluded that the ideal age is before six and other studies conclude after six [65,66]. For example, the study published by Lee et al. [60] concluded that FO should be provided to children younger than six. In their study 66 children from 1 to 12 years of age, showed that the greatest changes in the RPCS were for preschool-age children (under 7), and that children older than 7 presented a minimal correction. It could be concluded that the younger the patient is, the greater the possibility to correct the PFF. However, it should also be noted that natural foot development occurs before 6–7 years of age [11,24,67]. Moreover, it is not yet known whether gender is an etiological factor. Some papers indicate that gender influences the prevalence of PFF, showing a higher incidence in male children [25,31,68,69], which agrees with the present study as most of the included participants were male (60.7% boys).

Another of the etiological factors related to PFF is a low level of physical activity [11,17,70,71]. We planned to collect information about participants physical activity, but those data were not provided by any author. Regarding pain levels, only three of the included studies data related to this [42,43,55], where an improvement of pain after FO use was shown [35,46].

Previous studies showed great confusion in what to call the present pathology (i.e., flat foot valgus, pes planus, etc.) [13,71]. However, that was not an issue in the present study as the final diagnosis of all the authors was PFF, differentiating between asymptomatic or symptomatic patients.

The Beighton scale, navicular drop and RPCS tests, and X-rays were the most widely used tests by the included authors to assess PFF. Other tests, such as the arch height index, pedigraphy, Jack´s test, double/single heel rise test or FPI, were also used but not unanimously. Only the FPI has been validated for children under 6 years old. However, they show great specificity and sensitivity in adults [25,27,72,73,74]. The use of radiographs for the diagnosis of PFF is considered the “gold standard”. However, due to all the ethical problems caused by radiation, and the fact that an accurate diagnosis can be reached with clinical tests, radiographs are not used daily for the diagnosis of PFF.

Some of the tests have highlighted great controversy because the same values were not used by the different authors. For example, for Jafarnezhadgero et al. [4,54] the navicular drop was considered positive when the result was greater than 10 mm; however, Hsieh et al. [55] considered a positive result when it was greater than or equal to 6 mm. Given this ambiguity, it is not surprising that treatments such as FO do not have scientific evidence. After previous reviews and meta-analyses, it is difficult to obtain clear results. Morrison et al. [46] surveyed podiatrists, orthotists and physiotherapists in the United Kingdom about PFF diagnoses and came to the conclusion that what podiatrists used the most was the heel rise test, FPI and joint mobility to diagnose PFF. Recently, Zukauskas et al. [75] indicated that the navicular drop, FPI and CSI should be used for children between 5–8 years of age.

This systematic review presents some strengths; for example, the included studies presented a high level of evidence and a large number of scientific databases were reviewed. The main limitation of the present study is the small number of the included studies and participants, which could reduce the external validity of these results. Even though numerous studies related to FO and PFF are available in the literature, we have only found seven studies that were published in the last five years with a good methodological quality. Another limitation was the diversity of the outcome measures used and the heterogeneity of the interventions. Most of the studies were carried out on the Asian continent. The ethnic characteristics of each population are different, and these could have influenced the development and results of the treatment.

Future research should be undertaken with standardised diagnostic protocols with validated tests. These studies should be performed with a larger sample size and a longer-term follow-up (more than 3 months). In addition, studies should be separated between those who include children younger than 6 years old and those who include participants older than 6 years old. Finally, studies which assess the whole participant, including general painful symptoms and quality of life should be conducted. Then, the evidence for the effectiveness of FO treatment would be more concrete.

## 5. Conclusions

Conclusions from this review should be viewed with caution due to the low number of the included studies. The best type of FO and the optimal time of use cannot be concluded due to the heterogeneity between studies. There is no algorithm for PFF diagnosis, and there is a great diversity of clinical tests, characteristic signs and symptoms and radiographic measurements for PFF diagnosis. There is no universally accepted definition for PFF, although all authors of the included studies define it when there are more than two characteristic signs and symptoms or positive tests. The use of FO with high medial longitudinal arch may improve the signs and symptoms in some patients with PFF.

## Figures and Tables

**Figure 1 children-10-00371-f001:**
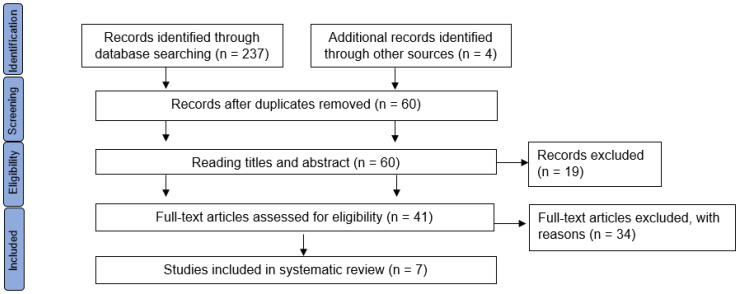
Article selection flowchart. Adapted from Preferred Reporting Items for Systematic Reviews and Meta-Analyses (PRISMA).

**Figure 2 children-10-00371-f002:**
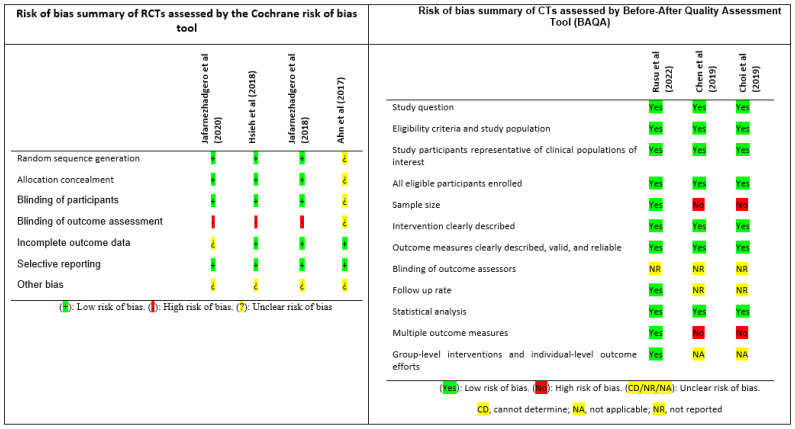
Risk of bias of the included studies. RCT: randomised controlled trial; CCT: controlled clinical trial; BAQA: Before-After Quality Assessment Tool [4,42,43,53,54,55,56].

**Table 1 children-10-00371-t001:** Study characteristics and publication characteristics.

Evidence Level by Sign	Author, Country (Year of Publication); Study Design	Sample Size	Year of Age (Mean/SD); Gender (M/F)	Weight in kg (SD); Height in cm (SD)	BMI kg/m^2^ (SD)	Previous Treatment	Diagnosis, Symptoms
II+A	Rusu et al., Romania (2022) [53]; RCT	CG: 15	9.37 (1.42); (17/13)	41.8 (12.72); 148.7 (10.96)	18.84 (5.32)	N/A	Bilateral PFF level II, asymptomatic
		IG:15					
II+A	Jafarnezhadgero et al., Iran (2020) [4]; RCT (single blind)	CG: 15	CG: 10.4 (1.5); (15/0)	CG: 48.2 (5.4); 141.2 (6.1)	CG: 20.1 (4.2)	N/A	PFF
		IG:15	IG: 10.5 (1.4); (15/0)	IG: 48.1 (9.1); 142.4 (5.7)	IG:20.0 (4.0)		
II+A	Chen et al., Taiwan (2019) [42]; CT	CG: 343	CG: 4.4 (7.9) meses; (187/156)	CG:18.0 (3.4); 104.5 (6.7)	CG:16.4 (2)	No	PFF, symptomatic
		IG: 123	IG: 4.3 (11.2); (77/46)	IG:18.2 (4.0); 105 (7.6)	IG:16.4 (2.1)		
II+B	Choi et al., South Korea (2019) [43]; CT	IG:18	IG:10.22 (0.43); (10/8)	N/A	N/A	N/A	PFF, symptomatic
		CG:13	CG:10.15 (0.38); (9/4)				
II+B	Hsieh et al., Taiwan (2018) [55]; RCT (single blind)	IG: 26	IG: 6.9 (0.6); (14/12)	N/A	N/A	N/A	PFF, symptomatic
		CG:26	CG: 6.2 (0.4); (14/12)				
II+A	Jafarnezhadgero et al., Iran (2018) [54]; RCT (single blind)	CG: 15	CG: 10.4 (1.5); (15/0)	GC: 48.2 (5.4); 141.2 (6.1)	CG: 20.1 (4.2)	N/A	PFF, N/A
		IG:15	IG: 10.5 (1.4); (15/0)	GE: 48.1 (9.1); 142.4 (5.7)	IG:20 (4)		
I+B	Ahn et al., South Korea (2017) [56]; CT	IG: 20	CG: 10.4 (4.99); (12/8)	GC:35.13 (16.93); 138.23 (10.17)	CG:18.37 (4.67)	N/A	PFF, N/A
		CG:20	IG: 9.59 (4.24); (12/8)	GE:37.41 (11.33); (139.28 ±12.78)	IG:19.18 (2.39)		

SD: standard deviation; M: male; F: female; Kg: kilogram; Cm: centimetre; BMI: body mass index; RCT: randomised controlled trial; CT: controlled trial; CG: control group; IG: intervention group; SIGN: Scottish Intercollegiate Guidelines Network; N/A: not applicable; PFF: paediatric flexible flatfoot.

**Table 2 children-10-00371-t002:** Sample selection and diagnoses.

Sample Selection			Diagnoses
Authors	Inclusion Criteria	Exclusion Criteria	PFF Definition and Assessment
Rusu et al. (2022) [53]	PFF after static and dynamic analysis	Surgery of foot or ankle; lower limbs pain; overweight; neuromuscular or neurological disorders.	Static and dynamic analysis: Clinical examination in standing and walking. Arch height index and subtalar flexibility, which was assessed by a force platform
Jafarnezhadgero et al. (2020) [4]	Navicular Drop > 10mm, RCPS > 4º eversion, Navicular Height < 0.31	Surgery or fracture of foot or ankle; neuromuscular problems, asymmetry of > 5mm.	Static and dynamic analysis: collapsed LMA while standing position and recovered when offloading. Navicular drop> 10mm. RCPS: >4º de eversión. Navicular height: <0.31.
Chen et al. (2019) [42]	PFF symptoms	Musculoskeletal injury, neurological disorder, previous FO	Anamnesis, Beighton scale, static and dynamic analysis: foot pain, fatigue and instability during walking, and changes in the normal morphology of the foot. Pedigraphy: + (CSI > 62.7 %)
Choi et al. (2019) [43]	PFF	Systematic inflammatory disease, lower limb trauma or surgery affecting their alignment.	Anamnesis: Characteristic signs and symptoms of PFF. Double/Single Heel Rise Test: + Test de Windlas: + X-ray: loadbearing, anteroposterior and lateral projections of the rearfoot (Saltzman and el-Khoury).
Hsieh et al. (2018) [55]	Symptomatic PFF (foot or calf pain, fatigue when walking or gait disturbances)	Surgery of foot or ankle; lower limb abnormalities, neuromuscular or neurological disorders.	Beighton scale: >4 Navicular drop: ≥ 6 mm FPI: >6 X-Ray: loadbearing, anteroposterior and lateral projections
Jafarnezhadgero et al. (2018) [54]	Boys from 8 to 12 years. Navicular Drop > 10mm, RCPS > 4º eversion, Navicular Height < 0.31	Surgery or fracture of foot or ankle; neuromuscular problems, asymmetry of > 10mm.	Static and dynamic analysis: collapsed LMA while standing position and recovered when offloading. Navicular drop > 10mm. RCPS: >4º de eversión. Navicular height: <0.31.
Ahn et al. (2017) [56]	PFF	Rigid FF, hereditary or neuromuscular diseases, fixed foot deformity or surgery of foot or ankle	RCPS: >4º de eversión. X-Ray: loadbearing, anteroposterior and lateral projections

mm: millimetre; RCPS: relaxed calcaneus position in standing; PFF: paediatric flexible flatfoot; FO: foot orthoses; CSI: Chippaux–Smirak index; FPI: foot posture index; FF: flat foot; MLA: medial longitudinal arch.

**Table 3 children-10-00371-t003:** Intervention characteristics.

Authors	Intervention	FO Material	FO Use	Education	Treatment Duration
Rusu et al. (2022) [53]	CG (n = 15): workout	Semirigid thermoplastic	Daily	FO use and normal BADL	3 months
	IG (n = 15): Personalised FO with LMA, heel cup and metatarsal dome + workout				
Jafarnezhadgero et al. (2020) [4]	CG (n = 15): placebo FO	GC: Polyester resin	Daily	Progressive FO use. Footwear New Balance 749, USA	4 months
	IG (n = 15): FO with LMA	GE: EVA			
Chen et al. (2019) [42]	CG (n = 343): none	Polipropilene and EVA	Daily	N/A	Mean of 11, 3 months
	IG (n = 123): FO with LMA				
Choi et al. (2019) [43]	IG (n = 18): FO with LMA	EVA, plastazote, poron, evazote or ucolite.	Daily	FO use and replace FO every 6 months	3–6 years
	CG (n = 13): none				
Hsieh et al. (2018) [55]	IG (n = 26): FO with LMA	Thermoplastic and EVA	Daily, minimum 5 h	FO use and comfortable footwear	3 months
	CG (n = 26): none				
Jafarnezhadgero et al. (2018) [54]	IG (n = 15): FO with LMA	GC: Polyester resin	Daily	Progressive FO use. Footwear New Balance 749, USA	4 months
	CG (n = 15): placebo FO	GE: EVA			
Ahn et al. (2017) [56]	IG (n = 20): inverted Blake’s FO with LMA	N/A	Daily, minimum 8 h	FO use	12 months
	CG (n = 20): inverted Blake’s FO				

CG: control group; IG: intervention group; FO: foot orthoses; LMA: longitudinal medial arch; BADL: Basic Activities of Daily Living, mm: millimetre; EVA: ethylene-vinyl acetate; N/A: not applicable.

## Data Availability

No new data were created or analysed in this study. Data sharing is not applicable to this article.

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
