# Peer review of "Efficacy of Plantar Orthoses in Paediatric Flexible Flatfoot: A Five-Year Systematic Review"

_children, 2023, doi:10.3390/children10020371_

Round 1

Reviewer 1 Report

Good review, complete and interesting

We don't suggest any modification

Author Response

R1

Comments and Suggestions for Authors

Good review, complete and interesting

We don't suggest any modification

Thank you very much.

Reviewer 2 Report

Dear Authors,

Building an meta-analysis for foot insole use in pediatric flat foot it is a subject of great interest for the readers.

Some major revisions are needed until publication:

- why did you choose to review only the last 5 years?

Reviewing only the last 5 years will throw out a lot of quality articles and studies that were done previously and has precious data.

We all know that in the 80s there was a trend that every flat foot should be treated with foot insoles. But in the 90s and 2000s a lot of publications showed that insoles are not a cure for the flat foot, and does not modify the final outcome (were a lot of studies that followed study and control groups that showed no modification for the pediatric flat foot at the end of the treatment). A proper meta-analysis should look further into the subject. Only by looking in the last years where there is a tendency to publish articles with bias risk does not make a good metaanalysis.

For example, study built by Rusu et al. [51] had only 30 participants in total, from wich 15 were in each study group. 15 participants are at most "case series" and do now allow for proper statistics. They did not meet the minimum requirements for a powerful study.

4 out of 6 studies presented in Table 1 are with less then 30 participants per group.

More about the study of Rusu el al.: you have stated that you only choosed the studies that did x-ray to diagnose flat foot (standing AP and lateral) - line 218, but this study only did clinical diagnosis. Flat foot cannot be diagnosed only by clinical assessment. Radiological and clinical examination make the positive diagnosis.

"The included studies were published in the last 5 years and all of them were developed in the Asian continent" - hard to belive that only Asians have flat foot and good results by foot orthoses - risk of bias by not taking into account the other parts of the world. Maybe some had worst response after foot insoles.

Your Conclusions "The use of FO with high medial longitudinal arch is an effective treatment to the 396 improve the signs and symptoms in the PFF" cannot be sustained by this small and inefficient meta analysis.

If this was true, than nobody whould have done surgeries for flat foot anymore. But, as other greater than us stated, such as Vincent Mosca, Paley or Evans, flat foot is a condition that needs apropriate tratment, and not all the time the answer will be foot insoled.

Author Response

R2

Comments and Suggestions for Authors

Dear Authors,

Building an meta-analysis for foot insole use in pediatric flat foot it is a subject of great interest for the readers.

Some major revisions are needed until publication:

Dear reviewer,

Thank you very much for giving us the possibility of addressing all the questions that arose during the review process. We think those comments have greatly improved the quality of this systematic review. Please find below all the responses in a point-by-point fashion. In the new revised version, the changes are highlighted in red font.

- why did you choose to review only the last 5 years?

Reviewing only the last 5 years will throw out a lot of quality articles and studies that were done previously and has precious data. We all know that in the 80s there was a trend that every flat foot should be treated with foot insoles. But in the 90s and 2000s a lot of publications showed that insoles are not a cure for the flat foot, and does not modify the final outcome (were a lot of studies that followed study and control groups that showed no modification for the pediatric flat foot at the end of the treatment). A proper meta-analysis should look further into the subject. Only by looking in the last years where there is a tendency to publish articles with bias risk does not make a good metaanalysis.

For example, study built by Rusu et al. [51] had only 30 participants in total, from wich 15 were in each study group. 15 participants are at most "case series" and do now allow for proper statistics. They did not meet the minimum requirements for a powerful study.

Thank you for your comment. As you have mentioned, in the 90s and 2000s, many publications concluded that foot orthoses (FO) were not a cure for the flat foot. However, the main limitation of those studies is that their objective was to assess structure changes of the foot, i.e. morphology, which did not occur, so they concluded that FO showed no modification to the pediatric flat foot (PFF). Nevertheless, the upward trend during the last years, is to focus their FO studies on outcomes such as kinetics and kinematics changes, muscle activity and gait.

The previous trend was to assess changes in the morphology of the foot, but it has been demonstrated that FO mainly affect the kinetics of the foot. Also, that FO can show a positive impact on a wide variety of outcomes such as pain, foot posture, gait or foot function in PFF. Therefore, the authors understood the necessity of carrying out a systematic review to demonstrate the efficacy of FO as a conservative treatment to reduce signs and symptoms in patients with PFF.

We considered the possibility of meta-analysis and planning to use standard statistical techniques. However, substantial heterogeneity did not occur between included studies. Therefore, we were not able to perform a meta-analysis.

4 out of 6 studies presented in Table 1 are with less then 30 participants per group.

The following sentence has been added to the limitation section: “The main limitation of the present study is the small number of the included studies and participants, which could reduce external validity of these results…”

More about the study of Rusu el al.: you have stated that you only choosed the studies that did x-ray to diagnose flat foot (standing AP and lateral) - line 218, but this study only did clinical diagnosis. Flat foot cannot be diagnosed only by clinical assessment. Radiological and clinical examination make the positive diagnosis.

Thank you for your comment. To carry out an x-ray to diagnose a flat foot was not an inclusion criteria of the present systematic review. The inclusion criteria were “The outcomes considered were those used to evaluate the improvement of signs and symptoms of the PFF”. Therefore, the study of Rusu el al. was included.

There are some tests used to confirm the PFF diagnosis without the necessity for an x-ray. For example the Foot Posture index (FPI), which has been mentioned in the study, is a validated tool for the measurement and diagnosis of a flat foot (Martínez-Nova A, Gijón-Noguerón G, Alfageme-García P, Montes-Alguacil J, Evans AM. Foot posture development in children aged 5 to11 years: A three-year prospective study. Gait Posture. 2018 May;62:280-284. doi: 10.1016/j.gaitpost.2018.03.032. Epub 2018 Mar 26. PMID: 29604617.).

"The included studies were published in the last 5 years and all of them were developed in the Asian continent" - hard to belive that only Asians have flat foot and good results by foot orthoses - risk of bias by not taking into account the other parts of the world. Maybe some had worst response after foot insoles.

We understand that fact can be a limitation of the present systematic review. This is why we decided to point it out in the limitation section in the last version of the manuscript: “All the studies were carried out in the Asian continent”.

6 of the 7 included studies were carried out in the Asian continent (Iran, Korea and Taiwan), and the other included study, which is also the most recent one, was carried out in Europe (Romania). We did not establish any inclusion/exclusion criteria in terms of the location of the study, nor in terms of studies with positive conclusions towards FO.

Your Conclusions "The use of FO with high medial longitudinal arch is an effective treatment to the 396 improve the signs and symptoms in the PFF" cannot be sustained by this small and inefficient meta analysis.

If this was true, than nobody whould have done surgeries for flat foot anymore. But, as other greater than us stated, such as Vincent Mosca, Paley or Evans, flat foot is a condition that needs apropriate tratment, and not all the time the answer will be foot insoled.

Thank you for your comment. However, the present study is a systematic review, we did not carry out a meta-analysis due to heterogeneity of included studies.

There are previous systematic reviews focused on FO in PFF, but they differ from the present systematic review because: they included any type of study (rather than randomised controlled trials as the present manuscript), they included very old studies, they carried out the search only in a few databases and their search strategies were brief. All these factors, among others, could explain the difference with respect to this systematic review and the previous ones, and consequently the presented conclusions.

The following sentence has been added to the conclusion section: “However, conclusions from this review should be viewed with caution due to the low number of the included studies”.

Reviewer 3 Report

Dear Author, thank you for the opportunity to review this article.

The introduction is thorough and suffices in theory about the clinical and radiological assessment of FFF. Besides the Beighton scale, one can also use the Wynne-Davies criteria which mostly sembles with the aforementioned.

Materials and Methods are adequate. Why did you exclude so many full-text articles? In row 303, what was the placebo for FO?

In line 90 you could also mention the most common surgical options for persistent symptomatic flatfoot, such as subtalar arthroereisis. Here is an article you can use as a reference: The Role of Arthroereisis in Improving Sports Performance, Foot Aesthetics and Quality of Life in Children and Adolescents with Flexible Flatfoot, published in Children, DOI 10.3390/children9070973.

Author Response

R3

Comments and Suggestions for Authors

Dear Author, thank you for the opportunity to review this article.

Dear reviewer,

Thank you very much for giving us the possibility of addressing all the questions that arose during the review process. We think those comments have greatly improved the quality of this systematic review. Please find below all the responses in a point-by-point fashion. In the new revised version, the changes are highlighted in red font.

The introduction is thorough and suffices in theory about the clinical and radiological assessment of FFF. Besides the Beighton scale, one can also use the Wynne-Davies criteria which mostly sembles with the aforementioned.

Thank you for your suggestion. When the authors were conceiving and designing the study, both the Beighton scale and Wynne-Davies criteria were considered. However, the included studies only use the Beighton scale, hence we considered only mentioning the Beighton scale in our manuscript.

The following information and reference have been added: “… flexibility assessment (most commonly assessed by Beighton scale or the Wynne-Davies criteria [23]), etc. 

  1. Wynne-Davies, R. Acetabular dysplasia and familial joint laxity: two etiological factors in congenital dislocation of the hip. A review of 589 patients and their families. J. Bone Joint Surg. Br. 1970, 52, 704–716.

Materials and Methods are adequate. Why did you exclude so many full-text articles? In row 303, what was the placebo for FO?

The main reason to exclude the studies were as follow: no randomised controlled trials, studies which did not include two groups (one of the groups being the intervention group with foot orthoses and the other group a control group) and poor methodological quality.

The process was as follows:

A total of 241 studies were identified from the electronic databases, reduced to 60 after duplications were removed. These were screened by title and abstract, and then the remaining 41-full-text articles were assessed for eligibility. Finally, 7 studies were included.

The placebo FO that are mentioned in row 303 are described in table 3. For the authors Jafarnezhadgero et al., a placebo FO included a 2mm flat FO without corrective elements. However, for the authors Ahn et al., the control FO was a Blake's inverted FO without high medial longitudinal arch and without corrections in the rearfoot.

In line 90 you could also mention the most common surgical options for persistent symptomatic flatfoot, such as subtalar arthroereisis. Here is an article you can use as a reference: The Role of Arthroereisis in Improving Sports Performance, Foot Aesthetics and Quality of Life in Children and Adolescents with Flexible Flatfoot, published in Children, DOI 10.3390/children9070973.

Thank you for your suggestion. The recommended study is now referenced in the manuscript and the following sentence has been added (lines 88): “Surgery, including procedures such as subtalar arthroereisis [30], is reserved for feet that have severe deformity, rigid FF, or FF with persistent symptoms that do not improve with conservative treatment”.

Round 2

Reviewer 2 Report

Revision reply:

Dear Authors,

I have read your revision. Here are my replies to your comments:

Q: - why did you choose to review only the last 5 years?

Reviewing only the last 5 years will throw out a lot of quality articles and studies that were done previously and has precious data. We all know that in the 80s there was a trend that every flat foot should be treated with foot insoles. But in the 90s and 2000s a lot of publications showed that insoles are not a cure for the flat foot, and does not modify the final outcome (were a lot of studies that followed study and control groups that showed no modification for the pediatric flat foot at the end of the treatment). A proper meta-analysis should look further into the subject. Only by looking in the last years where there is a tendency to publish articles with bias risk does not make a good metaanalysis.

For example, study built by Rusu et al. [51] had only 30 participants in total, from wich 15 were in each study group. 15 participants are at most "case series" and do now allow for proper statistics. They did not meet the minimum requirements for a powerful study.

RE: Thank you for your comment. As you have mentioned, in the 90s and 2000s, many publications concluded that foot orthoses (FO) were not a cure for the flat foot. However, the main limitation of those studies is that their objective was to assess structure changes of the foot, i.e. morphology, which did not occur, so they concluded that FO showed no modification to the pediatric flat foot (PFF). Nevertheless, the upward trend during the last years, is to focus their FO studies on outcomes such as kinetics and kinematics changes, muscle activity and gait.

The previous trend was to assess changes in the morphology of the foot, but it has been demonstrated that FO mainly affect the kinetics of the foot. Also, that FO can show a positive impact on a wide variety of outcomes such as pain, foot posture, gait or foot function in PFF. Therefore, the authors understood the necessity of carrying out a systematic review to demonstrate the efficacy of FO as a conservative treatment to reduce signs and symptoms in patients with PFF.

We considered the possibility of meta-analysis and planning to use standard statistical techniques. However, substantial heterogeneity did not occur between included studies. Therefore, we were not able to perform a meta-analysis.

RE: RE: article title should be changed accordingly. “a systematic review” with “2017-2022 systematic review” or “a 5 year systematic review”. In the need to obtain comprehensive results, one would take your study for a long term systematic review. Changing the title is advised.

Q: 4 out of 6 studies presented in Table 1 are with less then 30 participants per group.

RE: The following sentence has been added to the limitation section: “The main limitation of the present study is the small number of the included studies and participants, which could reduce external validity of these results…”

RE: RE: Ok

Q: More about the study of Rusu el al.: you have stated that you only choosed the studies that did x-ray to diagnose flat foot (standing AP and lateral) - line 218, but this study only did clinical diagnosis. Flat foot cannot be diagnosed only by clinical assessment. Radiological and clinical examination make the positive diagnosis.

RE: Thank you for your comment. To carry out an x-ray to diagnose a flat foot was not an inclusion criteria of the present systematic review. The inclusion criteria were “The outcomes considered were those used to evaluate the improvement of signs and symptoms of the PFF”. Therefore, the study of Rusu el al. was included.

There are some tests used to confirm the PFF diagnosis without the necessity for an x-ray. For example the Foot Posture index (FPI), which has been mentioned in the study, is a validated tool for the measurement and diagnosis of a flat foot (Martínez-Nova A, Gijón-Noguerón G, Alfageme-García P, Montes-Alguacil J, Evans AM. Foot posture development in children aged 5 to11 years: A three-year prospective study. Gait Posture. 2018 May;62:280-284. doi: 10.1016/j.gaitpost.2018.03.032. Epub 2018 Mar 26. PMID: 29604617.).

RE:RE: FPI it is an additional clinical test that can pe performed. Has it been performed in all of the selected studies from the systematic review? Please include in Table 1 how the diagnosis for PFF was done for each study (clinical / clinical + standing AP and lateral / or other method and state the one used).

Q: "The included studies were published in the last 5 years and all of them were developed in the Asian continent" - hard to belive that only Asians have flat foot and good results by foot orthoses - risk of bias by not taking into account the other parts of the world. Maybe some had worst response after foot insoles.

RE: We understand that fact can be a limitation of the present systematic review. This is why we decided to point it out in the limitation section in the last version of the manuscript: “All the studies were carried out in the Asian continent”.

6 of the 7 included studies were carried out in the Asian continent (Iran, Korea and Taiwan), and the other included study, which is also the most recent one, was carried out in Europe (Romania). We did not establish any inclusion/exclusion criteria in terms of the location of the study, nor in terms of studies with positive conclusions towards FO.

RE: RE: Ok

Q: Your Conclusions "The use of FO with high medial longitudinal arch is an effective treatment to the 396 improve the signs and symptoms in the PFF" cannot be sustained by this small and inefficient meta analysis.

If this was true, than nobody whould have done surgeries for flat foot anymore. But, as other greater than us stated, such as Vincent Mosca, Paley or Evans, flat foot is a condition that needs apropriate tratment, and not all the time the answer will be foot insoled.

RE: Thank you for your comment. However, the present study is a systematic review, we did not carry out a meta-analysis due to heterogeneity of included studies.

There are previous systematic reviews focused on FO in PFF, but they differ from the present systematic review because: they included any type of study (rather than randomised controlled trials as the present manuscript), they included very old studies, they carried out the search only in a few databases and their search strategies were brief. All these factors, among others, could explain the difference with respect to this systematic review and the previous ones, and consequently the presented conclusions.

The following sentence has been added to the conclusion section: “However, conclusions from this review should be viewed with caution due to the low number of the included studies”.

RE: RE: The phrase “The use of FO with high medial longitudinal arch is an effective treatment to improve the signs and symptoms in the PFF” should be soften and combined with the sentence that you added.

Replace with “The use of FO with high medial longitudinal arch may improve the signs and symptoms in some patients with PFF.”, and add as last sentence.

Conclusions could look like:

Conclusions from this review should be viewed with caution due to the low number of the included studies. The best type of FO, and the optimal time of use, cannot be concluded due to the heterogeneity between studies. There is no algorithm for PFF diagnosis, and there is a great diversity of clinical tests, characteristical signs and symptoms, and radiographic measurements for PFF diagnosis. There is no universally accepted definition for PFF, although all authors of the included studies define it when there are more than two characteristical signs and symptoms or positive tests. The use of FO with high medial longitudinal arch may improve the signs and symptoms in some patients with PFF.

Author Response

Dear Authors,

I have read your revision. Here are my replies to your comments:

Dear reviewer,

Thank you very much for giving us the possibility of addressing all the questions that arose during the review process. We think those comments have greatly improved the quality of this systematic review. Please find below all the responses in a point-by-point fashion. In the new revised version, the changes are highlighted in red font.

Q: - why did you choose to review only the last 5 years?

Reviewing only the last 5 years will throw out a lot of quality articles and studies that were done previously and has precious data. We all know that in the 80s there was a trend that every flat foot should be treated with foot insoles. But in the 90s and 2000s a lot of publications showed that insoles are not a cure for the flat foot, and does not modify the final outcome (were a lot of studies that followed study and control groups that showed no modification for the pediatric flat foot at the end of the treatment). A proper meta-analysis should look further into the subject. Only by looking in the last years where there is a tendency to publish articles with bias risk does not make a good metaanalysis.

For example, study built by Rusu et al. [51] had only 30 participants in total, from wich 15 were in each study group. 15 participants are at most "case series" and do now allow for proper statistics. They did not meet the minimum requirements for a powerful study.

RE: Thank you for your comment. As you have mentioned, in the 90s and 2000s, many publications concluded that foot orthoses (FO) were not a cure for the flat foot. However, the main limitation of those studies is that their objective was to assess structure changes of the foot, i.e. morphology, which did not occur, so they concluded that FO showed no modification to the pediatric flat foot (PFF). Nevertheless, the upward trend during the last years, is to focus their FO studies on outcomes such as kinetics and kinematics changes, muscle activity and gait.

The previous trend was to assess changes in the morphology of the foot, but it has been demonstrated that FO mainly affect the kinetics of the foot. Also, that FO can show a positive impact on a wide variety of outcomes such as pain, foot posture, gait or foot function in PFF. Therefore, the authors understood the necessity of carrying out a systematic review to demonstrate the efficacy of FO as a conservative treatment to reduce signs and symptoms in patients with PFF.

We considered the possibility of meta-analysis and planning to use standard statistical techniques. However, substantial heterogeneity did not occur between included studies. Therefore, we were not able to perform a meta-analysis.

RE: RE: article title should be changed accordingly. “a systematic review” with “2017-2022 systematic review” or “a 5 year systematic review”. In the need to obtain comprehensive results, one would take your study for a long term systematic review. Changing the title is advised.

Thank you for your suggestion. The title has been changed: “Efficacy of plantar orthoses in paediatric flexible flatfoot: a 5 year systematic review”.

Q: 4 out of 6 studies presented in Table 1 are with less then 30 participants per group.

RE: The following sentence has been added to the limitation section: “The main limitation of the present study is the small number of the included studies and participants, which could reduce external validity of these results…”

RE: RE: Ok

Thank you. 

Q: More about the study of Rusu el al.: you have stated that you only choosed the studies that did x-ray to diagnose flat foot (standing AP and lateral) - line 218, but this study only did clinical diagnosis. Flat foot cannot be diagnosed only by clinical assessment. Radiological and clinical examination make the positive diagnosis.

RE: Thank you for your comment. To carry out an x-ray to diagnose a flat foot was not an inclusion criteria of the present systematic review. The inclusion criteria were “The outcomes considered were those used to evaluate the improvement of signs and symptoms of the PFF”. Therefore, the study of Rusu el al. was included.

There are some tests used to confirm the PFF diagnosis without the necessity for an x-ray. For example the Foot Posture index (FPI), which has been mentioned in the study, is a validated tool for the measurement and diagnosis of a flat foot (Martínez-Nova A, Gijón-Noguerón G, Alfageme-García P, Montes-Alguacil J, Evans AM. Foot posture development in children aged 5 to11 years: A three-year prospective study. Gait Posture. 2018 May;62:280-284. doi: 10.1016/j.gaitpost.2018.03.032. Epub 2018 Mar 26. PMID: 29604617.).

RE:RE: FPI it is an additional clinical test that can pe performed. Has it been performed in all of the selected studies from the systematic review? Please include in Table 1 how the diagnosis for PFF was done for each study (clinical / clinical + standing AP and lateral / or other method and state the one used).

Assessment information is available in Table 2; however, the column has been improved because more detailed information has been added.

Each of the included authors defined PFF depending on different assessments. All authors included some physical assessment, however, only 3 of the included authors (Choi et al, Hsieh et al and Ahn et al) also included radiographic assessment. On the other hand, only Chen et al and Choi et al pointed out that determined signs and symptoms (i.e., foot pain, fatigue and instability during walking) are required to diagnose PFF. Jafarnezhadgero et al diagnoses of PFF were defined when patients presented with collapsed medial longitudinal arch (MLA) while in a standing position, which recovered when offloading, navicular Drop > 10mm, RCPS >4º eversion and Navicular Height < 0,31. Finally, Rusu et al. defined a PFF diagnoses based on the arch height index and the subtalar flexibility, which was assessed using a force platform.

Q: "The included studies were published in the last 5 years and all of them were developed in the Asian continent" - hard to belive that only Asians have flat foot and good results by foot orthoses - risk of bias by not taking into account the other parts of the world. Maybe some had worst response after foot insoles.

RE: We understand that fact can be a limitation of the present systematic review. This is why we decided to point it out in the limitation section in the last version of the manuscript: “All the studies were carried out in the Asian continent”.

6 of the 7 included studies were carried out in the Asian continent (Iran, Korea and Taiwan), and the other included study, which is also the most recent one, was carried out in Europe (Romania). We did not establish any inclusion/exclusion criteria in terms of the location of the study, nor in terms of studies with positive conclusions towards FO.

RE: RE: Ok

Thank you. 

Q: Your Conclusions "The use of FO with high medial longitudinal arch is an effective treatment to the 396 improve the signs and symptoms in the PFF" cannot be sustained by this small and inefficient meta analysis.

If this was true, than nobody whould have done surgeries for flat foot anymore. But, as other greater than us stated, such as Vincent Mosca, Paley or Evans, flat foot is a condition that needs apropriate tratment, and not all the time the answer will be foot insoled.

RE: Thank you for your comment. However, the present study is a systematic review, we did not carry out a meta-analysis due to heterogeneity of included studies.

There are previous systematic reviews focused on FO in PFF, but they differ from the present systematic review because: they included any type of study (rather than randomised controlled trials as the present manuscript), they included very old studies, they carried out the search only in a few databases and their search strategies were brief. All these factors, among others, could explain the difference with respect to this systematic review and the previous ones, and consequently the presented conclusions.

The following sentence has been added to the conclusion section: “However, conclusions from this review should be viewed with caution due to the low number of the included studies”.

RE: RE: The phrase “The use of FO with high medial longitudinal arch is an effective treatment to improve the signs and symptoms in the PFF” should be soften and combined with the sentence that you added.

Replace with “The use of FO with high medial longitudinal arch may improve the signs and symptoms in some patients with PFF.”, and add as last sentence.

Conclusions could look like:

Conclusions from this review should be viewed with caution due to the low number of the included studies. The best type of FO, and the optimal time of use, cannot be concluded due to the heterogeneity between studies. There is no algorithm for PFF diagnosis, and there is a great diversity of clinical tests, characteristical signs and symptoms, and radiographic measurements for PFF diagnosis. There is no universally accepted definition for PFF, although all authors of the included studies define it when there are more than two characteristical signs and symptoms or positive tests. The use of FO with high medial longitudinal arch may improve the signs and symptoms in some patients with PFF.

Thank you, the conclusion has been modified as suggested.

Reviewer 3 Report

The paper is ready to be publish

Author Response

The paper is ready to be publish

Thank you very much.